# Assessment of the Geomorphological Heritage of the Costa Branca Area, a Potential Geopark in Brazil

**Marco Túlio Mendonça Diniz** [1,*] , **Maria Luiza de Oliveira Terto** [2] **and Fernando Eduardo Borges da Silva** [2,*]

1    Department of Geography, Seridó Higher Education Centre, Federal University of Rio Grande do Norte, Caicó 59300-000, Brazil
2    Centre for Humanities, Literature and Arts, Federal University of Rio Grande do Norte, Natal 59078-970, Brazil
*    Correspondence: tuliogeografia@gmail.com (M.T.M.D.); fernando100borges00.1@gmail.com (F.E.B.d.S.)

**Abstract:** The Atlantic margin of South America is considered passive and stable. However, there are some local points of discordance within the Brazilian coastal region with more than 7490 km of extension, where there is only one tectonic relief. These sites are located in a marginal sedimentary basin in which there is a small area of Quaternary tectonic relief, which makes it scientifically relevant at an international level. The present work proposes using two methods for quantifying the geomorphological heritage of this area. The main difference between the methods is the use of aesthetic values together with scientific ones as central values in one of the methods, while the other method focuses only on scientific values. The quantitative evaluation performed here allowed for the identification of seven geomorphosites with one method and only four with the other. Considering the results obtained, meetings were held with civil society and with the state and local municipalities which presented the possibility of creating a geopark, given the area's importance for understanding the history of the Earth and potential as a priority area for geoconservation.

**Keywords:** geodiversity; geomorphological heritage; geodiversity assessment

## 1. Introduction

Appearing as a topic of interest as late as the 1990s, geodiversity currently occupies a prominent space in the geosciences [1] and has been receiving increasing attention in recent years. The concept of geodiversity was formulated by Gray [2], who defined it as "the natural range (diversity) of geological (rocks, minerals, fossils), geomorphological (relief forms, topography, physical processes), soil and hydrological features. It includes their assemblages, structures, systems, and contributions to landscapes".

The understanding of geoheritage arises from the need for territorial knowledge, specifically from the issue tangent to the selection of the most relevant places concerning the natural abiotic heritage, its consequent valuation as such, and hierarchisation before other similar ones. This need is a result of debates on geopatrimonial values which have been discussed by several authors [1–6] and which support the notion of natural abiotic heritage, namely the geoheritage.

With this clear heritage perspective, the geoheritage would therefore be an asset consisting of tangible or intangible values. The term geoheritage designates the heritage granted to this and future generations by the evolution of the planet Earth, which is worthy of appreciation and conservation [7]. It corresponds to an asset belonging to a culture, a country, or even humankind, and requiring special care. Geoconservation seeks the preservation of geodiversity and attempts to maintain the geopatrimony for future generations.

The procedures for the selection of sites of particular heritage interest have been studied by several authors [1–6,8–11] with a certain procedural consensus [1]. These have included the inventory (qualitative evaluation) and quantification steps which are essential

for the construction of databases that allow the analysis of a given territory, the selection of sites, and a classification of their respective hierarchies. This information allows the suggestion of geoconservation and geotourism measures.

Thus, the objective of this work is to employ two distinct methodological proposals to quantify the geopatrimony of the only area of tectonic relief on the Brazilian coast, specifically in the municipalities of Grossos, Areia Branca, and Porto do Mangue in the state of Rio Grande do Norte, Northeast Brazil.

Aesthetic values are of paramount importance for some of the primary objectives of geoconservation because the aesthetic dimensions of a site are often configured as the main attraction for tourism [12]. The aesthetic elements of an area are also central to proposals for nature conservation, as highlighted by the UNESCO concept of natural heritage, which made "Natural monuments consisting of physical and biological formations or groups of such formations of outstanding universal value from the aesthetic or scientific point of view".

The recognition of the need to valorise the geopatrimony of an area through the popularisation of knowledge about the Earth's history led to UNESCO's geoparks program, which aims to encourage local and international efforts for the conservation of abiotic resources. Conceptually, the term "geopark" has been attributed to areas with special geological characteristics, associated with abiotic resources, which can contribute to sustainable local development, thus providing opportunities and promoting enthusiasm for geoconservation, as they become new sites of tourist attraction [13].

In this work, we used the criteria presented by Vlachopoulos and Voudouris [13] to propose the potential new global UNESCO geopark in Serifos, Greece; the authors considered that the proposed geopark was located in an area that encapsulated features of special geological significance, rarity, or beauty. The potential Costa Branca geopark is situated in an area comprising extremely rare reliefs and remarkable scenic beauty. The region has a special geological and geomorphological significance and, therefore, we propose the creation of a global UNESCO geopark within it.

## 2. Methods

Two quantification procedures were followed in this study, the first of which was developed by Emmanuel Reynard [8–10]. Considering that some sites of high aesthetic value may not have a high scientific value, an evaluation was also made based on a second method developed by Diniz, Araújo, and Chagas [11]. The latter emphasises that aesthetic values, as well as scientific ones, can help identify areas of representative geomorphological interest called geomorphosites. As defined by Panizza [14], a geomorphosite is "a landform to which a value can be attributed".

The studied sites were inventoried and subsequently quantified according to Emmanuel Reynard's quantification method established in [8–10], as well as the method established by Diniz, Araújo, and Chagas [11]. Then, analyses and comparisons between the methodologies were performed and pertinent discussions were held.

### 2.1. Study Area

The Atlantic margin of the South American Continent is considered passive, with an extension of more than 11,000 km from Guyana to Argentina. Most of this stretch belongs to Brazil, which has around 7,500 km of coastline in the passive margin. However, a punctual stretch of this margin hosted Quaternary tectonic activity which generated cliff uplift in rocks dating from the Neogene in the Ponta do Mel region. Neotectonics have been recorded in other stretches of the Brazilian coast [15], but these were unable to generate "tectonic reliefs" on the coast.

Within this margin, a stretch of the Brazilian coast extending over 70 km has potential for the creation of a geopark; the area is located entirely in the "Costa Branca" region and represents the coastline delimited by [16] Icapuí (state of Ceará) in the NW and Touros (state of Rio Grande do Norte) in the SE. The area receives this denomination,

which in English means "White Coast", from the high salinity of the water and lands, with characteristics favourable for the production of sea salt (Figure 1).

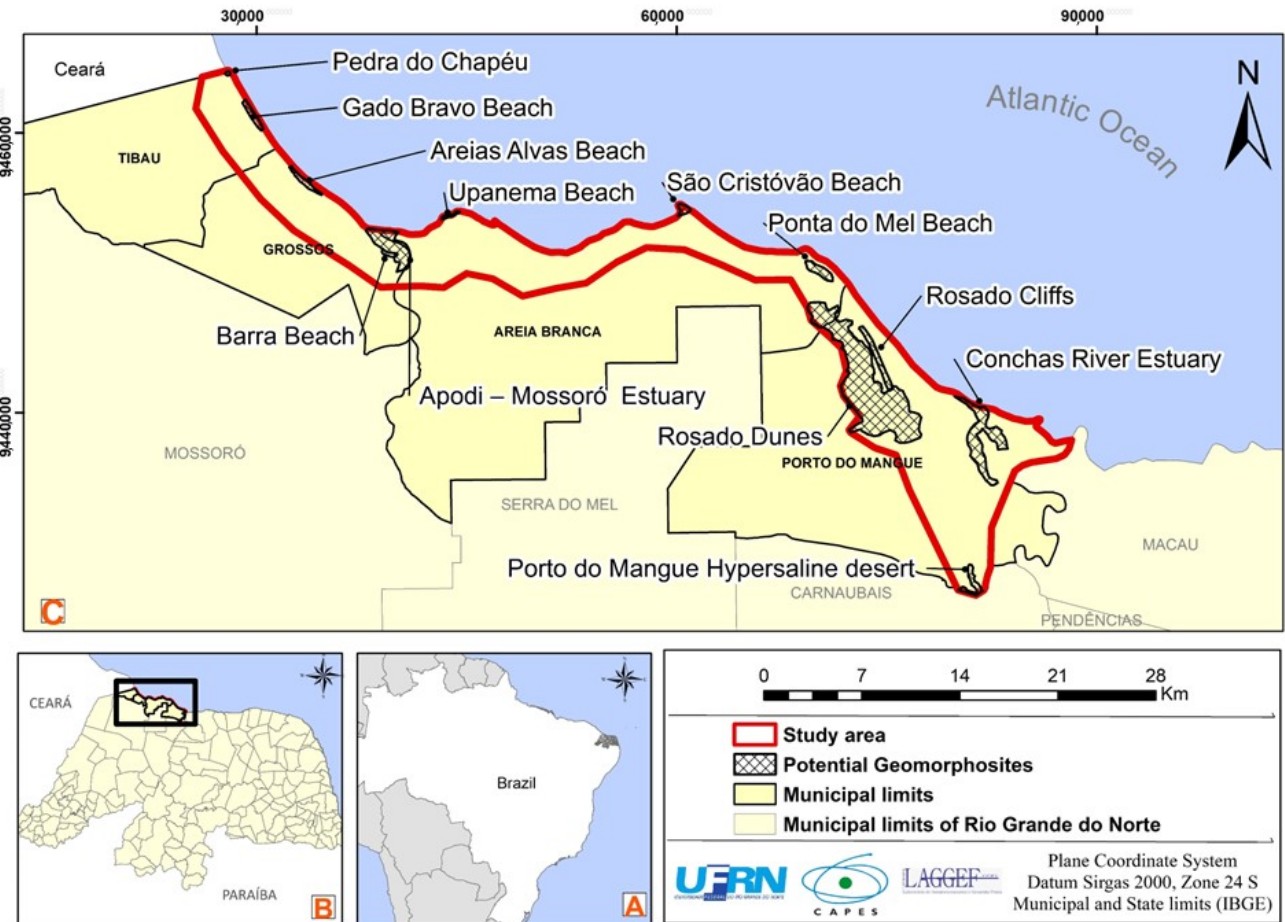

**Figure 1.** (**A**) Map showing the location of the geosites in the study area, (**B**) location of the area in the state of Rio Grande do Norte and (**C**) location of the state of Rio Grande do Norte in South America. Source: elaborated by the authors.

Both the high salinity of the area and the presence of white dunes are attributes favoured by the semi-arid climate characteristic of this coast. Notably, the climate of the region is hot and dry, with a long dry season and irregular precipitation; the rains are concentrated between the months of February to May [17], with an average rainfall of less than 750 mm/year [17]. The area is the driest stretch of the entire Brazilian coast, another peculiar characteristic of the area.

The Polo Costa Branca was proposed in 1995 by the Secretary of Tourism of the state of Rio Grande do Norte (SETUR—RN) in order to promote tourism in the region, becoming in 2004 the Costa Branca Tourist Region by decree n.18.187/05. The area has a vast tourist potential due to the landscapes within it, highlighting the municipalities of Tibau, Grossos, Areia Branca, and Porto do Mangue, covered in this study.

The area is situated within the Potiguar Basin, a marginal sedimentary basin that originated from the deposition of sediments in the open rift in the Borborema Province as a consequence of the rupture of the Gondwana supercontinent. Sedimentation in the Potiguar Basin began before the opening of the Atlantic rift during the lower Cretaceous. However, its most superior formations, such as the Jandaíra Formation, are marine and date back to the Upper Cretaceous, i.e., after the opening of the Atlantic. Its emerging portion consists of asymmetric grabens separated by high basement and limited by

two platforms, with sedimentary filling linked to transgressive marine deposits of Albian to Campanian age [18] (Figure 2).

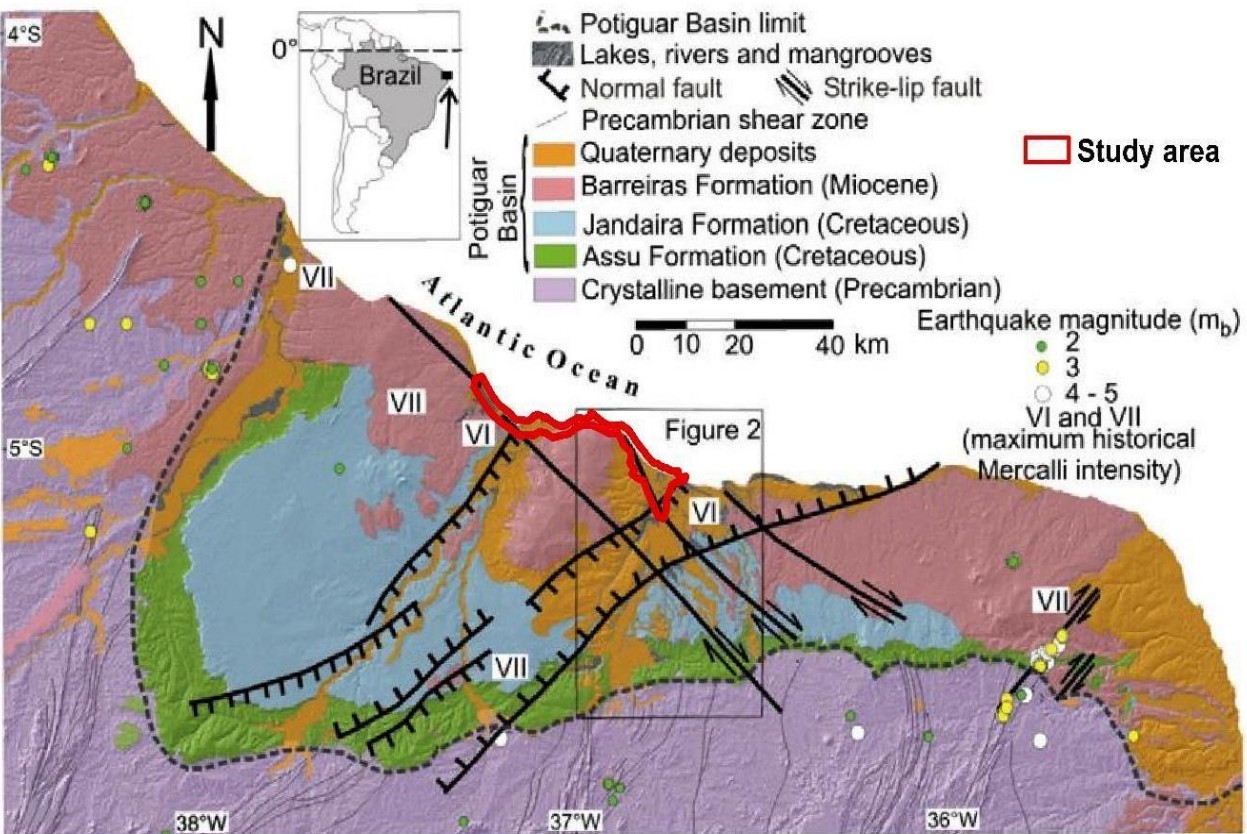

**Figure 2.** Map showing the location of the Potiguar Basin, its main formations, and the location of the study area. Source: modified from Angelim et al. [19] and Moura Lima et al. [20].

Among the sedimentary formations in the area is the Assu Formation (94–89 Ma) which is characterised by proximal siliciclastic deposits. According to Moura-Lima et al. [20] the sediments of the Açu Formation are overlain by sediments belonging to the Jandaíra Formation (90–70 Ma), and the two are the main Mesozoic formations of the Potiguar Basin.

The Tibau Formation (82–72 Ma) comprises thick clastics overlying the Guamaré carbonates which consist of thick hyaline sandstones that formed within a coastal fan depositional environment; these can be seen to outcrop in the study area in the form of an active cliff on the Tibau/RN beach. These formations could not be mapped on the scale shown in the map in Figure 2.

The Barreiras Formation (23–1,6 Ma) is characterised by a Cenozoic tabular relief with sea limits in the form of cliffs. The formation overlies Precambrian igneous, metamorphic, and sedimentary rocks in outcrops on the sublittoral coastal portion. Subsequent to the Barreiras formation, as a result of the tectonic inversion, occurs the emergence of the Serra do Mel dome structure (5,3 Ma) [21].

According to Maia and Bezerra [22], compressional inversion structures controlled by faults developed during the Quaternary to form the Serra do Mel dome. The geomorphologic expression of the dome's uplift is exteriorised in the Ponta do Mel cliffs, which reach an altitude of about 120 m in Ponta do Mel (the central area of the dome reaches an altitude of 250 m). This gives a singularity to the cliffs in the area; being, therefore, relief formed by Quaternary tectonics, they are single exemple in the whole coast of the passive margin of South America [23].

Considering the main natural elements of the region, the potential geomorphosites of the study area were distinguished as: Pedra do Chapéu, Barra Beach, Areias Alvas Beach, Upanema Beach, Apodi-Mossoró Estuary, São Cristóvão Beach, Ponta do Mel, Porto do Mangue Hypersaline Desert, Rosado Cliffs, Rosado Dunes, and Conchas River Estuary.

The Pedra do Chapéu (hat stone) site corresponds to Cenozoic outcrops belonging to the Tibau Formation, consisting of medium and coarse sandstones with a yellow and greenish colouration and levels of clay with colour variations associated with pareidolia, in the shape of a hat. The features of this 10 m high cliff generate significant interest from tourists. The site has access via paved streets and there is no difficulty in reaching the site (Figure 3A).

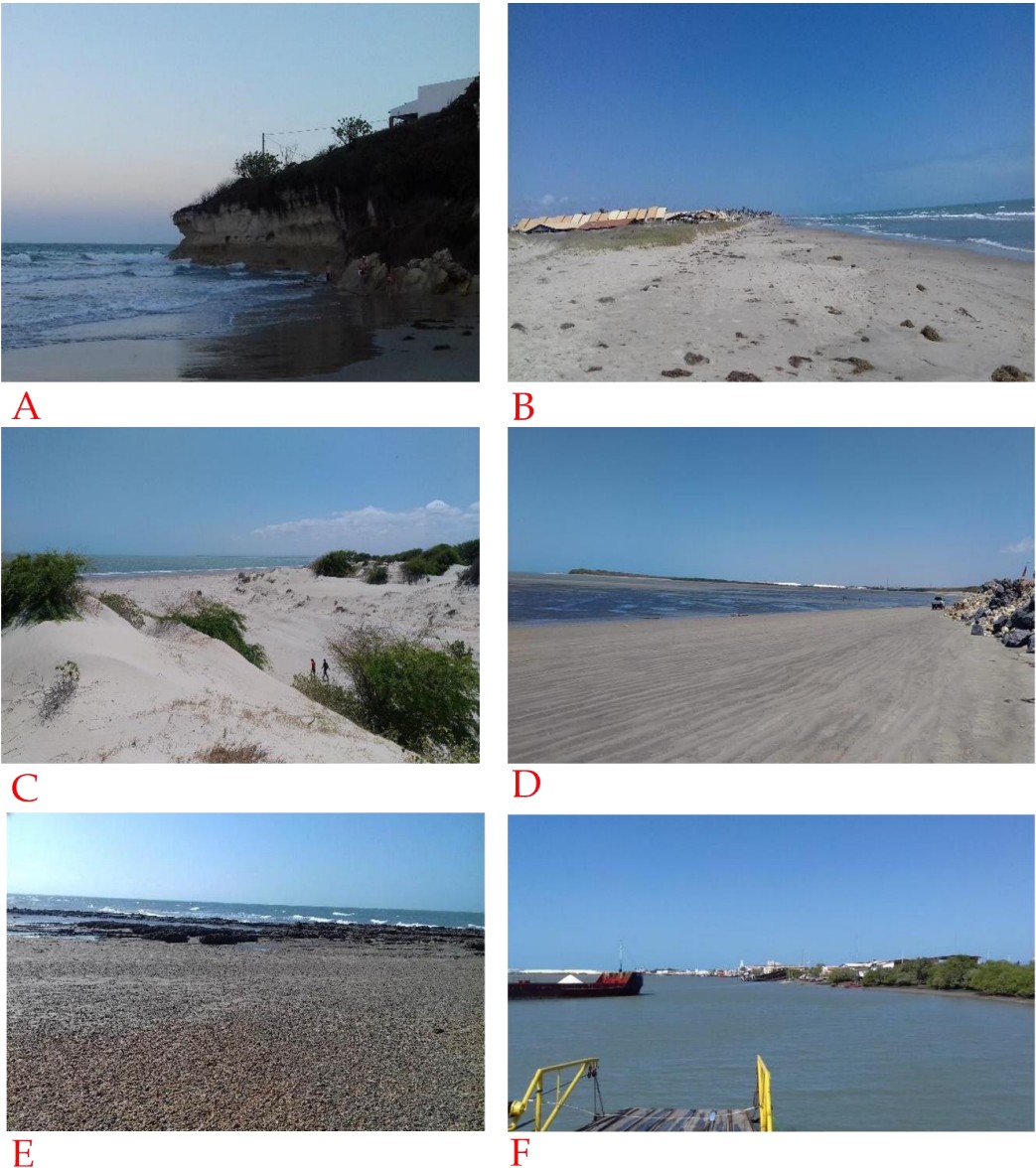

**Figure 3.** Photographs of the potential geomorphosites, namely Pedra do Chapéu (**A**); Gado Bravo Beach, with representative anthropisation (**B**); Areias Alvas Beach (**C**); Barra Beach (**D**); Upanema Beach (**E**); and the Apodi- Mossoró Estuary (**F**). Source: the authors.

The Gado Bravo Beach (Figure 3B) corresponds to a beach geoform characterized by intense anthropisation, with the densification of the coastal strip interfering with the beach morphology. Here, the local population is confronted with the consequences of the wind transport of sediments that reach the houses.

The Areias Alvas Beach (Figure 3C), located in the municipality of Grossos, corresponds to a beach with sandy dune deposits, moderate accessibility, satisfactory observation conditions, and excellent conservation status.

The Barra Beach (Figure 3D) hosts semi-fixed dunes and has shrubby vegetation of the Caatinga Biome characteristic of the interior of the region, which is a steppe savanna that predominantly occupies the portion of Brazil with a semi-arid tropical climate.

The Upanema Beach, located in Areia Branca, is a steep stretch of beach (extending over more than 100 m) comprising beach sandstones and records of Holocene sedimentary rock formations (beach rocks) (Figure 3E).

The Apodi-Mossoró River Estuary is located between the municipalities of Grossos and Areia Branca, and hosts Holocene deposits with marine and fluvial sedimentation. It is characterised, besides its economic use, by a strong connection with religious activities due to the population that lives on the banks of the estuary (Figure 3F).

The São Cristóvão Beach comprises cliffs and semi-fixed dunes occupying a canyon area with inactive erosive activity. It has good conditions of access and visualisation of features. The beach runs along a stretch characterised by minor tectonic activity that has uplifted the sandstones of the Barreiras Formation up to about 10 m height into active cliffs (Figure 4). The entire stretch of the Costa Branca consists of a low-lying coast with Quaternary sedimentation of marine terraces, beaches, and estuaries; the cliffs at the edge of the basin in Icapuí, state of Ceará, and the cliffs of São Cristóvão and Rosado are an exception because these localities have hosted enough tectonic activity to expose the Barreiras Formation, dating back to the Miocene [20] (Figures 4–6).

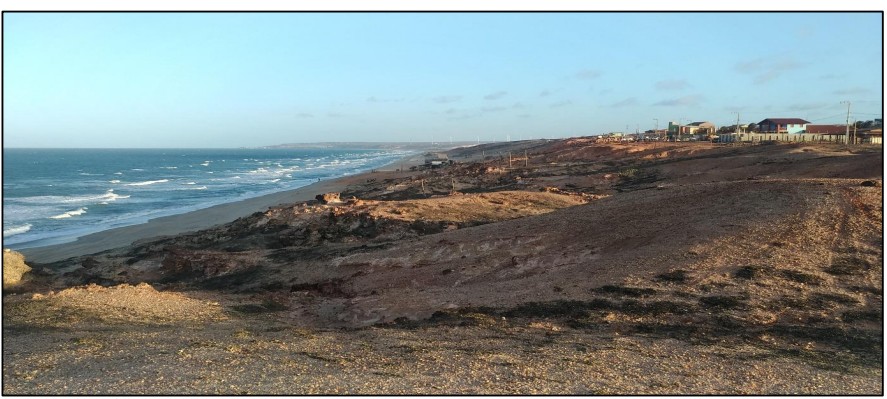

**Figure 4.** Cliffs of São Cristóvão. Source: the authors.

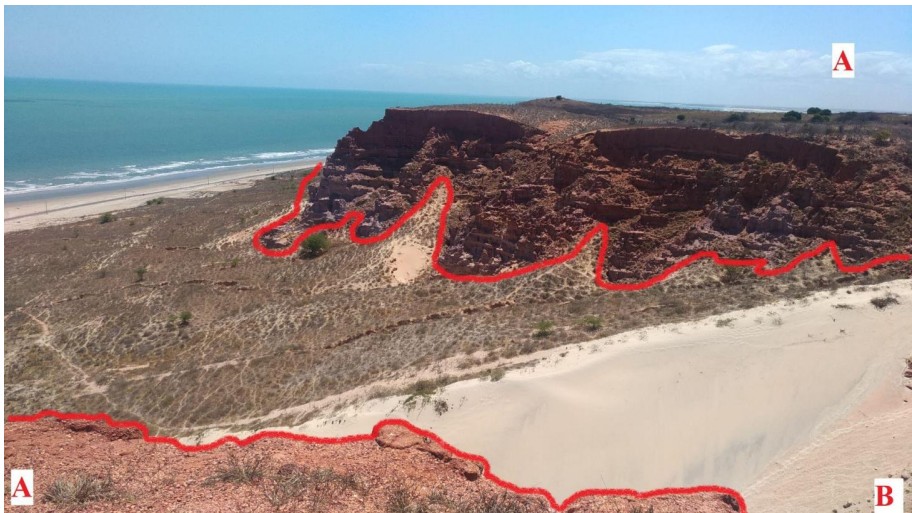

**Figure 5.** Features of Ponta do Mel. Legend: "A" inactive cliffs; "B" semifixed dunes. Source: the authors.

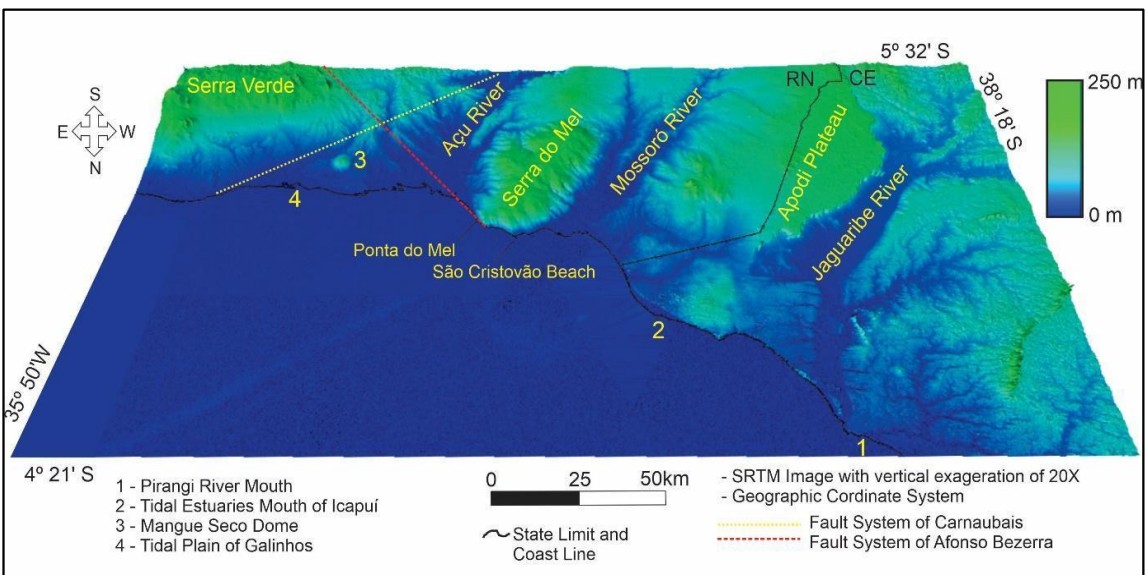

**Figure 6.** Digital figure elevation model of the Açu, Mossoró, and Jaguaribe estuaries. Source: Reprinted/adapted with permission from Ref. [24]. Copyright 2017, Diniz and Vasconcelos.

Ponta do Mel (Figure 5) has inactive cliffs with dune formations in the foothills as a result of wind action associated with low amounts of rainfall and beach rocks on the beach geoform. The stretch of coast adjacent to the cliffs of Ponta do Mel is the only one with tectonic relief on the entire coast of Brazil. Indeed, with the exception of the northern and southernmost countries of South America, which have been influenced by Andean tectonics, the Ponta do Mel area is the only one area with tectonic relief on the Atlantic passive margin of South America which runs from Guyana to Argentina.

The tectonics of the area are Quaternary in age. Maia and Bezerra [22] identified Quaternary sedimentation at the top of the Serra do Mel dome, and the Ponta do Mel cliff is the largest repercussion on the coast that promoted the uplift of the dome; the same occurred to a lesser extent in the active cliffs of the São Cristóvão Beach (Figure 6).

The Hypersaline Desert site in the municipality of Porto do Mangue (Figure 7) corresponds to the extensive and shallow fluviomarine plains of the Açu River. The soil impermeability, low and flat topography, and low flow and high evapotranspiration rates of the area prevent the formation of mangrove ecosystems, and the extreme local salinity inhibits the growth of most biotic species [24]. The salinity comes in from the tides which flood a large part of the plains because of their low altitude, generally below three metres. Due to the impermeability of the soils and the high evapotranspiration rates, the sodium chloride dissolved in the seawater will crystallise and salinise the soil, thus preventing the growth of biotic species and generating the desert landscape. The whitish and greyish colours in the area are the main characteristic of the White Coast (Figure 7).

The particularities and geomorphological features of the hypersaline desert are further influenced by the large extent of the fluviomarine plains, combined with strong winds and considerable sedimentary input. These in turn are responsible for the emergence of an embryonic dune field, formed by fine sandy sediments and fluvial muddy clays (Figure 7).

The site of the Rosado Cliffs (Figure 8) corresponds to low-height inactive cliff features situated parallel to the coastline and overlapping the Afonso Bezerra fault in the northwestern portion of the municipality of Porto do Mangue. In the highest portions, the cliff heights do not exceed 15 m, and their base is situated at an average altitude of 8 m. The cliffs consist of post-barrier alluvial fan deposits of Pleistocene age, the modelling of which is still occurring today through the action of wind, rain, and river flows due to the high friability of the material composing the cliffs [22].

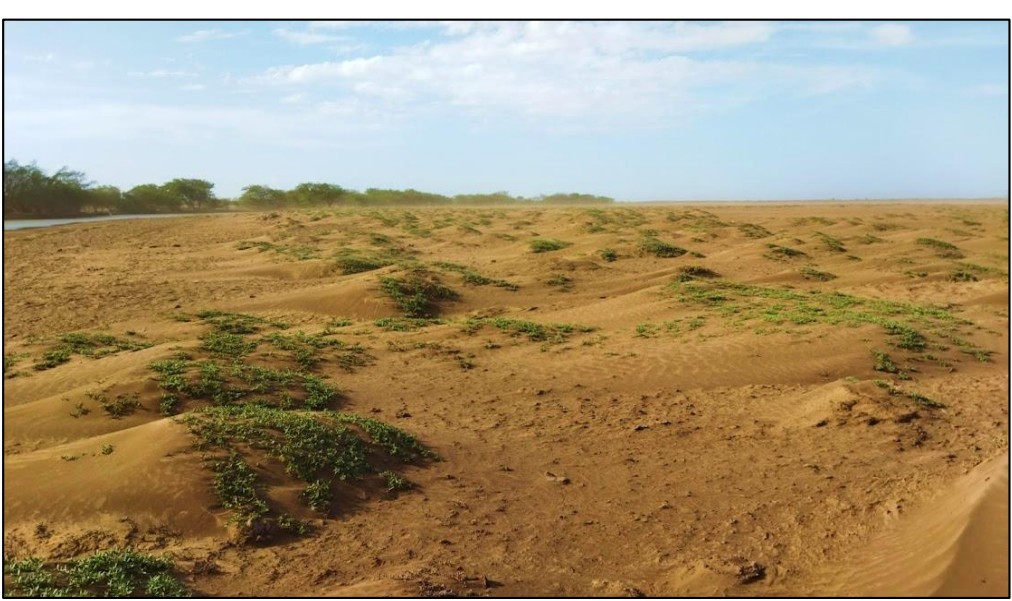

**Figure 7.** Hypersaline deserts near the main channel of the Açu River. Source: the authors.

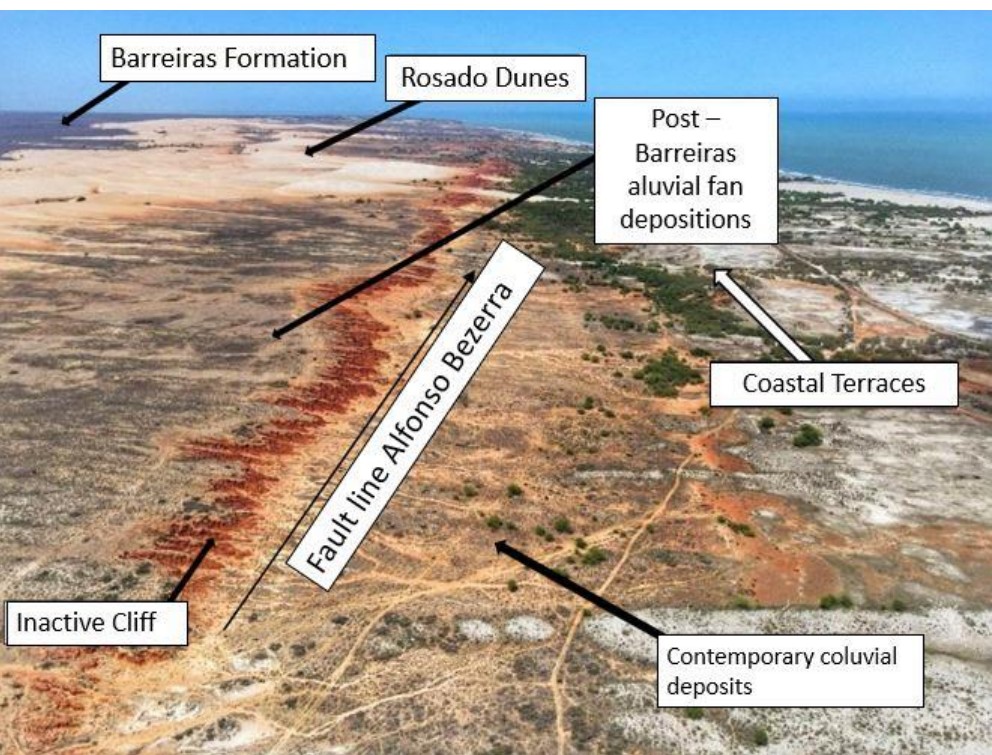

**Figure 8.** Aerial image of the Rosado Cliffs captured with a drone. Source: The authors.

There are no known examples of similar sites on the Brazilian coast and the Rosado Cliffs (Figure 8) are indeed the only example of tectonic relief of Holocene origin located on the Brazilian Atlantic margin. These cliffs would record the maximum of the marine transgression of about 2100 years past [25]. Given the continuity of the uplift process of these cliffs, the altitude measured at their base is discordant with the maximum transgression identified, this being also a repercussion of the Serra do Mel uplift process which, based on the evidence seems to follow active, is the main process responsible for the concomitant elevation of the cliffs (Figure 9).

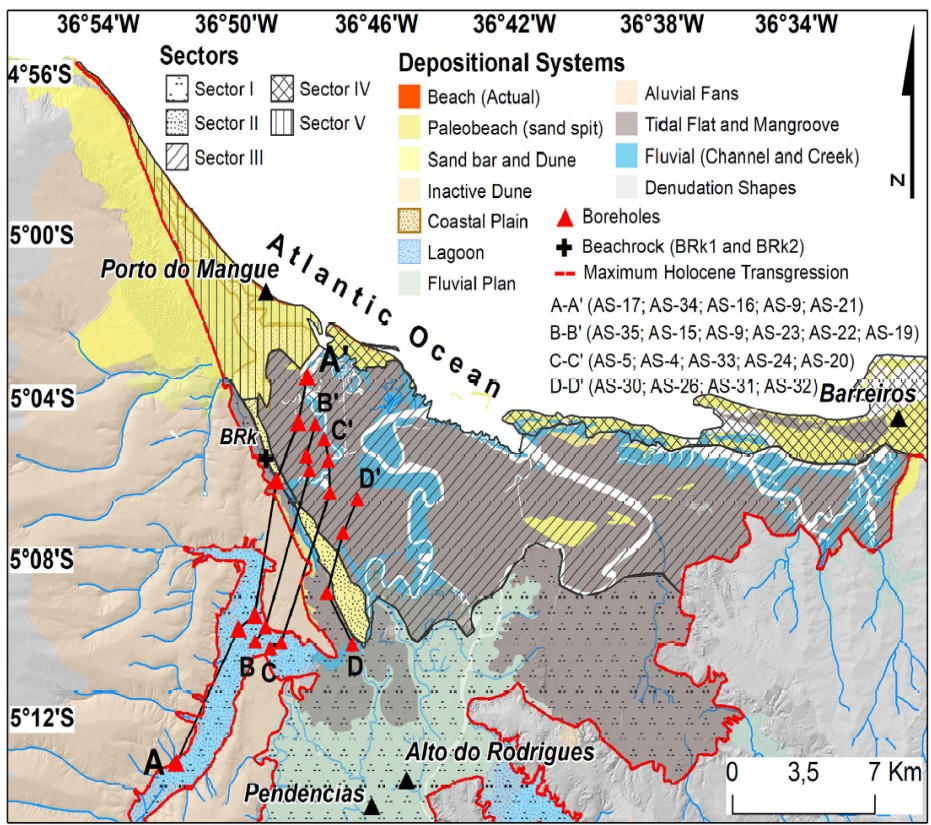

**Figure 9.** Main sedimentary sectors of the region. Source: Barbosa et al. [26].

The site of the Rosado Dunes is located between the municipalities of Porto do Mangue and Areia Branca and hosts the largest mobile dune field in the state of Rio Grande do Norte. The dunes are positioned on the leeward side of the Rosado Cliffs (Figure 10) and are characterised by a typical reddish colouration directly related to the Rosado Cliffs. The white sediments coming from the strand are mixed with the red sediments being eroded and transported from the cliffs, thereby giving the dunes a characteristic and unique colouration. There is no other known occurrence of mobile and active reddish dunes covering such a large area in Brazil, and, indeed, these landscapes are more common in Africa and on the Arabian Peninsula.

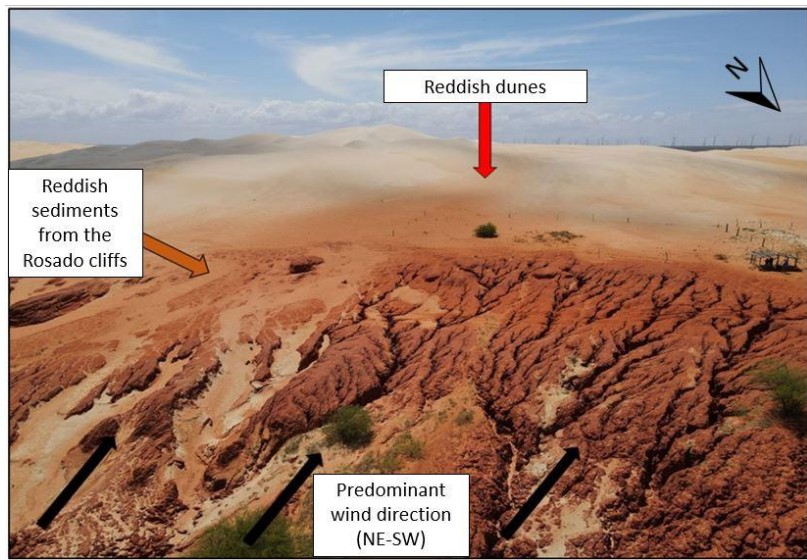

**Figure 10.** Natural dyeing process of the Rosado Dunes. Source: the authors.

The last site considered in the municipality of Porto do Mangue was the Conchas River Estuary, Figure 11, located in the urban core of the municipal seat. The estuary is part of the Açu River Delta, with the Conchas River bifurcating from the main channel of the Açu River about 12 km from the coast. The estuary is strongly anthropised and occupied by shrimp farming tanks and evaporation tanks used for the production of sea salt. The mangroves currently occur over a small stretch on the river margins. The city of Porto do Mangue is itself located in the estuary.

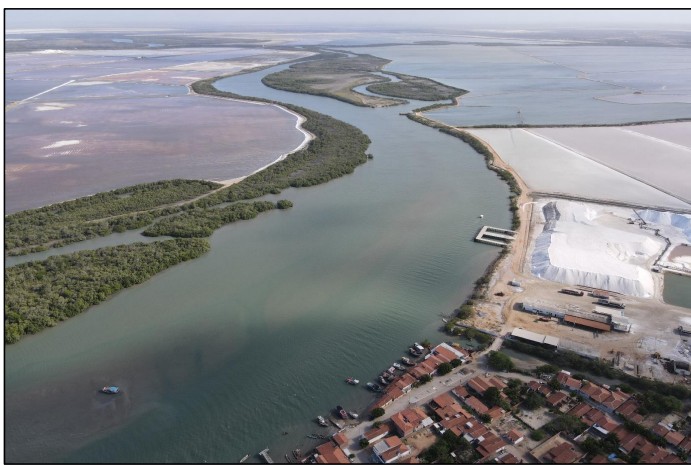

**Figure 11.** Mangrove, river channel, salt marshes, and the city of Porto do Mangue in the Conchas River Estuary. Source: The authors.

### 2.2. Methods

Two different methods were used for the quantitative evaluation. The first was developed by Emmanuel Reynard [8–10], in which a quantitative evaluation was carried out using an arithmetic mean for scientific, ecological, and aesthetic values and the cultural value criterion by the highest score. The scores assigned in each criterion corresponded to 0 (zero), 0.25; 0.5; 0.75; and 1. The scores equal to or greater than 0.75 were considered to represent high values (Table 1). The Reynard method considered a geomorphosite to be a site of interest with a high scientific value.

**Table 1.** Evaluation values and criteria of the Reynard method [8–10].

| Values | | Criteria | | Evaluation |
|---|---|---|---|---|
| Scientific value | Scientific value | | Integrity Representativeness Rareness | Informs about the state of conservation of the site. Level of exemplarity of the site. The rarity of the site as a reference space. |
| | | | Paleogeographic value | Importance of the site for the reconstruction of Earth or climate history. |
| | Ecological value | a. | Ecological impact | Importance of the site concerning a specific ecosystem and whether the site is already protected. |
| | | b. | Protect site | |
| Additional values | Aesthetic value | a. | Viewpoints | Potential for observation and scenic beauty. |
| | | b. | Contrasts, vertical development, and structure of space | |
| | Cultural value | a. | Religious importance | The performance of the site in a primary role in religious and/or spiritual terms; its historical context; its presence as inspiration for artists; relevance of the site in the history of Earth sciences. |
| | | b. | Historical importance | |
| | | c. | Artistic or literary importance | |
| | | d. | Geohistorical importance | |
| | | e. | Economic products | |

The second evaluation was based on an adaptation of methods presented by several authors, namely Pereira [6], who dealt with sustainable development and geoconservation in Chapada da Diamantina; Pereira [5] who studied the geomorphological heritage of the Montesino Natural Park; and Diniz, Araújo, and Chagas [11], who recently published a new method highlighting as core values aesthetic and scientific ones for the definition of geomorphosites.

According to the latter authors: "Aesthetic value has replaced intrinsic value, because aesthetics is an essential element of geomorphological heritage in attracting the attention of the public, as discussed by Coratza and Hobléa [27] It was thus decided to give it greater relevance, and it was proposed as a core value, alongside scientific value" [11]. The proposal used has so far been applied in three master's theses defended in Brazil [28–30].

The quantification of geomorphological heritage in this method was divided into three values: scientific, aesthetic, and touristic. In this proposal, a high score for the parameters of scientific and/or aesthetic value was needed to define a geomorphosite, bringing, as a differential, the aesthetic element as essential for the definition of areas of interest.

The scientific value was of fundamental importance for the definition of the areas of interest. Seven parameters were used to quantify this value, namely the degree of scientific knowledge, the didactic relevance and representativeness [6], the ecological interest [5], the paleogeographic value [8], and the diversity of the geomorphological aspects and ecodynamics of the environment [31] (Table 2). The scientific value (VCi) was calculated using Equation (1):

$$VCi = A1 + A2 + A3 + A4 + A5 + A6 + A7 \qquad (1)$$

**Table 2.** Parameters of scientific value.

| Scientific Value | | | | | | | |
|---|---|---|---|---|---|---|---|
| Parameters | A1—Degree of Scientific Knowledge | A2—Ecodynamic Units | A3—Representativeness of Geomorphological Materials and Processes | A4—Diversity of Geomorphological Aspects (shapes And Processes) | A5—Ecological Interest | A6—Paleogeographic Value | A7—Didactic Relevance |
| Definition | Indicates whether the geomorphosite itself has already been the subject of academic studies or been cited in technical-scientific papers. | Refers to the classification of the units at the highest taxonomic level. | Indicates the relevance of the geomorphosite as a record of elements or processes related to the geomorphological evolution of the region and the context in which it is inserted, as well as the use of geomorphology for society. | Elements of the geomorphology aggregated by geomorphosite. | Evaluates the relationship between the geomorphological object(s) and the occurrence of biological species ; the score increases with the perception of the relationship between habitats and geomorphology. | Assesses the importance of the object for the reconstruction of the history of the climate and the Earth (for example, providing a reference for a glacial stage). | Potential of the geomorphosite for illustrating elements or processes of geodiversity and the potential for using the site to teach geosciences and/or secondary schools. |
| 0 | No reference to the Geomorphosite. | Stable environment- the predominance of pedogenesis. Medium with a slow evolution, closed vegetation cover, moderate dissection, and absence of volcanic manifestations. | Absence of any relevant aspects of a scientific nature. | No geomorphological aspects. | No connection with biological elements. | No paleogeographic expressiveness. | No didactic relevance |
| 1 | Quoted in a technical report or monograph. | - | No potential. | One geomorphological aspect. | Fauna and/or flora of interest. | - | Likely to be used for didactic purposes post-graduation. |

**Table 2.** *Cont.*

| | | | | Scientific Value | | | |
|---|---|---|---|---|---|---|---|
| **Parameters** | **A1—Degree of Scientific Knowledge** | **A2—Ecodynamic Units** | **A3—Representativeness of Geomorphological Materials and Processes** | **A4—Diversity of Geomorphological Aspects (shapes And Processes)** | **A5—Ecological Interest** | **A6—Paleogeographic Value** | **A7—Didactic Relevance** |
| **2** | Cited in two monographs, scientific articles, or dissertations. | Intergrade (transition area for stability); when pedogenesis stands out over morphogenesis. | Involves illustrative records of elements or processes of geodiversity, but these have no potential. | Two geomorphological aspects. | One of the best places to observe fauna and/or flora of interest. | Contains illustrative elements, but with a difficult visualisation of paleographic elements. | Likely to be used for didactic purposes post-graduation. It can be used for didactic purposes for undergraduate students. |
| **3** | Cited in three theses, dissertations, or scientific papers. | Intergrade (transition area for instability); when morphogenesis stands out over pedogenesis. | Contains illustrative elements that represent sections of the type locality or are used as classic examples and anthropic interference. | Three geomorphological aspects. | The geomorphological characteristics condition the ecosystem(s). | Contains illustrative elements that represent the paleogeographic evolution of the area; it can be used as an example with good teaching resources and with human decharacterisation. | Can be used for teaching purposes for high school students. |
| **4** | Cited in >4 academic theses or articles in scientific journals. | Strongly unstable (predominance of morphogenesis); units with an intervention of geodynamics through volcanism, tectonic deformations, or anthropic instability. | Hosts illustrative elements that represent sections, type localities, or are used as classic examples of geomorphological elements or processes; a good resource for teaching, and/or the use of relief for the society. | Four or more geomorphological aspects. | The geomorphological characteristics determine the ecosystem(s). | Contains illustrative elements that represent the paleogeographic evolution of the area; it can be used as an example with good teaching resources and without the presence of mischaracterisation or vegetation cover; allows for an excellent visualisation of paleogeographic elements. | Can be used for teaching purposes for the general public or elementary school students. |

Source: Diniz, Araújo, and Chagas [11].

For the definition of the score, we considered as high values those that obtained a score higher than 75% of the total possible value, corresponding to scores from 22 to 28. Table 1 presents the categories evaluated in this criterion while Table 2 lists the scientific value parameters.

The aesthetic value has a central importance for the definition of areas of special geomorphological interest, being defined by the following criteria: rarity, integrity, variety of elements of geodiversity [6], visual quality, and observation conditions [4]. Thus, the aesthetic value (VEst) was calculated using Equation (2):

$$\text{Vest} = B1 + B2 + B3 + B4 + B5 \tag{2}$$

To be classed as having a high value, the site must achieve scores between 16 and 20 points (above 75% of the total value). According to Dupont, Antrop, and van Eetvelde [32], the criteria associated with aesthetic values are based on people's perceptions of landscapes. Indeed, the authors identified that individuals tended to turn their attention to areas with

open landscapes and contrasting elements such as colour differentiation and verticality. According to Kirillova [33], contrasts of colours and heterogeneous landscapes helped promote touristic interest. These criteria are described in Table 3.

**Table 3.** Parameters of aesthetic value.

| Parameters | B1—Rarity | B2—Integrity | B3—Variety of Elements of Geodiversity and/or Associated Themes | B4—Visual Quality | B5—Observation Conditions |
|---|---|---|---|---|---|
| **Aesthetic Value** | | | | | |
| **definition** | Importance of the site in terms of its geomorphological occurrence in the investigated area. | Indicates the level of conservation of the geomorphosites and the possibility of visualising the aspects of interest. | The number of interests and elements of geodiversity and themes associated with the geomorphological heritage (hydrology, hydrogeology, mineralogy, petrology, oceanography, hydrography, etc.). | The scenic beauty of the place. Measured by verticality, colour contrasts, and individual elements (inselbergs, ruiniform reliefs, etc.). | Conditions of visualisation of the elements of geodiversity. |
| **0** | Commonly occurring geomorphosite in the study area (more than ten occurrences within a 200 km radius). | Deteriorated and uncharacterised geomorphosite; the observation of the elements of interest is compromised and there is no possibility of recovery. | No association. | Geomorphosite with no aesthetic relevance. | No conditions for observation. |
| **1** | Between six and ten formations with similar characteristics in the area (within the same geomophological context and within a 200 km radius). | Deteriorated geomorphosites, but the visualisation of the aspects of interest is still possible, without the possibility of recovery. | Association with only one element or theme connected to geodiversity. | Geomorphosites occur in a pleasant place and with an individual element | Only visible with equipment. |
| **2** | Existence of up to five formations with similar characteristics in the area (within the same geomorphological context and within a 200 km radius). | Deteriorated geomorphosites; the visualisation of the aspects of interest is still possible, with the possibility of recovery. | Association with two elements or themes connected to geodiversity. | Geomorphosites occur in a pleasant place and have significant scenic appeal, with verticality (<50 m) or colour contrasts of four to six colours. | Limited by vegetation. |
| **3** | Existence of up to three formations with similar characteristics in the area (within the same geomorphological context and within a 200 km radius). | Geomorphosites with some deterioration, but the visualisation of the aspects of interest is still possible, with the possibility of recovery. | Association with three elements or themes connected to geodiversity. | Geomorphosites occur in a pleasant place and have significant scenic appeal, with verticality (>50 m), a mountainous relief, and contrasts of four to six colours. | Good, but only observable from the base. |
| **4** | Unique formation in the area within a 200 km radius or ≥3 within a 500 km radius. | Intact geomorphosites without any deterioration or need for recovery. | Association with more than four elements or themes connected to geodiversity. | Geomorphosites are characterised by aesthetic splendour and occur in a pleasant place and have significant scenic appeal. The areas have verticality (>50 m), a mountainous relief, and contrasts of seven colours or more. | Good, landscape with verticality which is visible from a scenic viewpoint. |

Source: Diniz, Araújo, and Chagas [11].

In addition, aesthetic criteria contribute greatly to geoconservation in areas of geomorphological interest which have not yet been examined in scientific studies. These areas would be valued with low score if they were evaluated only according to the scientific value as a core value, as proposed by existing methods [4,10]. We identified that areas with a high aesthetic value but little known scientific value due to the lack of research covering them were common in Brazil, considering the size of the territory relative to the amount of scientific research conducted.

The touristic value has a secondary importance and, according to Pereira [6], to achieve a high touristic value, an area must have ease of access (accessibility) through paved roads, for example; infrastructure that offers support to visitors near the geomorphosite; and visibility in tourist campaigns, especially those of national relevance. Besides the tourist categories, there are activities that can be developed in the locality [34] (Table 4).

**Table 4.** Parameters of touristic value.

| | | | Touristic Value | | |
|---|---|---|---|---|---|
| Parameters | C1—Accessibility | C2—Presence of Infrastructure | C3—Existence of Ongoing Use | C4—Scenery | C5—Tourism Category |
| Definition | Indicative of the difficulties accessing the site. | Indicative of the presence of infrastructure that facilitates and serves as support for visitors, such as the presence of bathrooms, tourist guides, accommodation (>3 km), restaurants (>3 km), and others. | Indicates the current conditions of tourist use of the geomorphosite. | Used in local/national/international tourism campaigns. | Existing types of tourism in the area (sun-and-beach, geotourism, cultural, religious, etc.) |
| 0 | Accessible from a >5 km-longtrail or through areas with containment works. | Absence of any Infrastructure. | Geomorphosite without any current use. | Does not appear in campaigns. | - |
| 1 | Accessible from a 2 to 5 km-long trail or through a private area. | Equipped with basic infrastructure, but which supports visitors, with the presence of one element. | Geomorphosite with some visitation which is still incipient. | Occasionally appears in local campaigns. | The site presents one type of tourism. |
| 2 | Accessible from unpaved roads or <2 km-long trails. | Equipped with basic infrastructure, but which supports visitors, with the presence of two elements. | Geomorphosite with an average visitation rate and the presence of accommodation. | Frequently appears in local campaigns. | The site features two types of tourism. |
| 3 | Accessible from paved roads or <2 km-long trails. | Equipped with basic infrastructure, but which supports visitors, with the presence of three elements. | Geomorphosite with a high visitation rate, without a visitor control mechanism, but with accommodation. | Occasionally appears in national campaigns. | The site features three types of tourism. |
| 4 | Accessible directly through main paved roads (federal, state, or municipal). | Equipped with full infrastructure that provides complete support for visitors, with the presence of four or more elements. | Geomorphosite with a high visitation rate and equipped with visitor control measures and accommodation facilities <3 km away. | Constantly seen in national campaigns. | The site features more than four types of tourism. |

Source: Diniz, Araújo, and Chagas [11].

To be classed as having a high touristic value, a site must reach a score between 16 and 20 points (above 75% of the total value). The touristic value (VTur) was calculated using Equation (3):

$$VTur = C1 + C2 + C3 + C4 + C5 \tag{3}$$

### 3. Results

The quantitative evaluation was carried out for the twelve potential sites of geomorphological interest distributed among the municipalities of Tibau, Grossos, Areia Branca, and Porto do Mangue. The knowledge of the sites was obtained through previous surveys and added field activity.

#### 3.1. Quantitative Evaluation of Geomorphodiversity Using the Reynard Method

The scientific value of the sites of interest was assessed against the criteria of integrity, representativeness, rarity, and paleogeographic value. The total value was obtained through the arithmetic mean of the scores obtained for each criterion, as shown in Table 5.

**Table 5.** Evaluation of the sites of interest according to the method of Reynard et al. [8,9] for scientific value.

| Places of Geomorphological Interest | Scientific Value | | | | | |
|---|---|---|---|---|---|---|
| | Integrity | Representativeness | Rarity | Geographic Paleo-Value | Total | Evaluation Classes |
| A—Pedra do Chapéu | 0.25 | 0.5 | 0.75 | 0.5 | 0.5 | Low |
| B—Gado Bravo Beach | 0 | 0.25 | 0 | 0 | 0.06 | Low |
| C—Areias Alvas Beach | 1 | 0.5 | 0.25 | 0.25 | 0.5 | Low |
| D—Barra Beach | 0.5 | 0.75 | 0.75 | 0.25 | 0.56 | Medium |
| E—Apodi-Mossoró Estuary | 0.25 | 0.5 | 0.5 | 1 | 0.56 | Medium |
| F—Upanema Beach | 0.75 | 0.75 | 0.5 | 0.75 | 0.69 | Medium |
| G—São Cristóvão Beach | 0.75 | 0.5 | 0.5 | 1 | 0.69 | Medium |
| H—Ponta do Mel Beach | 1 | 1 | 1 | 1 | 1 | **High** |
| I—Porto do Mangue Hypersaline Desert | 1 | 1 | 1 | 1 | 1 | **High** |
| J—Rosado Cliffs | 1 | 1 | 1 | 1 | 1 | **High** |
| K—Rosado Dunes | 1 | 1 | 1 | 0.25 | 0.81 | **High** |
| L—Conchas River Estuary | 0.75 | 1 | 0.25 | 0.25 | 0.56 | Medium |

Source: Reynard et al. [8,9].

According to the evaluation using the first method, the sites were considered to have a very low score of 0.25, a low score of 0.26 to 0.5, a medium score of 0.51 to 0.75, and a high score of 0.76 to 1. Ponta do Mel, the hypersaline desert of Porto do Mangue, and the Rosado Cliffs obtained the maximum score (1 point), followed by the Rosado Dunes with a score of 0.81; these were sites with a high scientific value. The beaches of Upanema, São Cristóvão, Barra, and the estuaries of Apodi-Mossoró and Conchas River received an average score. The other sites received low or very low scores, especially the Gado Bravo Beach, which had the lowest score of 0.06, due to the high degree of modifications to its natural features.

For the additional values, the highest scores were obtained for Ponta do Mel and the Apodi-Mossoró Estuary, while the São Cristóvão Beach, Pedra do Chapéu, and the other sites received average scores, as seen in Table 6. Since the additional values were not used in this proposal for the definition of geomorphosites, they were not ranked.

For the ecological value, identified through the arithmetic mean between the ecological impact and protected site criteria, the sites that achieved the highest scores were the Rosado Dunes and the Conchas River Estuary. Barra Beach and the Apodi-Mossoró River Estuary followed with 0.87 points. The Areias Alvas Beach and Ponta do Mel received 0.75 points, followed by the Porto do Mangue Hypersaline Desert and the other sites with low ecological values, namely Pedra do Chapéu and Gado Bravo Beach.

The aesthetic value was evaluated by the viewpoint, vertical development contrasts, and structure of the space criteria, and valued by the arithmetic mean of these elements. Among the most representative scores, Ponta do Mel, the Rosado Cliffs, and the Rosado Dunes scored highest, followed by Pedra do Chapéu (0.87) and the beaches of Upanema and São Cristóvão (0.75).

The cultural value of the sites was evaluated by their religious, historical, artistic/literary, geohistorical, and economic activities. The sites that received the highest

cultural score were the Apodi-Mossoró Estuary and Ponta do Mel, for being representative in the cultural, religious, and artistic elements.

**Table 6.** Evaluation of the sites of interest according to Reynard [8,9] for the additional values.

| | | | **Additional Values** | | | | |
|---|---|---|---|---|---|---|---|
| | | | | **Cultural Value** | | | |
| **SITES** | **V. Eco** | **V. Aest** | **Religious** | **History** | **Artistic/Literary** | **Geohistorical** | **Economic** |
| A—Pedra do Chapéu | 0.12 | 0.87 | 0 | 0.5 | 0.25 | 0.25 | 0.75 |
| B—Gado Bravo Beach | 0.12 | 0 | 0 | 0.25 | 0 | 0 | 0.25 |
| C—Areias Alvas Beach | 0.75 | 0.12 | 0 | 0 | 0 | 0.25 | 0 |
| D—Barra Beach | 0.87 | 0.12 | 0 | 0 | 0.25 | 0.25 | 0.5 |
| E—Apodi-Mossoró Estuary | 0.87 | 0 | 1 | 0.75 | 0.75 | 1 | 1 |
| F—Upanema Beach | 0.5 | 0.75 | 0 | 0 | 0.25 | 0.5 | 0.75 |
| G—São Cristóvão Beach | 0.5 | 0.75 | 0.5 | 0.25 | 0.25 | 0.75 | 0.5 |
| H—Ponta do Mel Beach | 0.75 | 1 | 1 | 0.75 | 1 | 1 | 0.75 |
| I—Porto do Mangue Hypersaline Desert | 0.62 | 0.37 | 0 | 0 | 0 | 0 | 1 |
| J—Rosado Cliffs | 0.37 | 1 | 0 | 0 | 0 | 1 | 0 |
| K—Rosado Dunes | 1 | 1 | 0 | 0 | 0.75 | 0.25 | 0 |
| L—Conchas River Estuary | 1 | 0.62 | 0.25 | 0.25 | 0 | 0.25 | 1 |

Source: Reynard et al. [8,9].

### 3.2. Quantitative Evaluation of Geomorphodiversity Using the Diniz, Araújo, and Chagas Method [11]

In the method proposed by Diniz, Araújo, and Chagas [11], only those sites that achieved a high score for their scientific and/or aesthetic value (central values) were considered geomorphosites. Those that obtained a medium score for the central criteria and/or scored high on the additional criteria were considered geodiversity sites.

For the scientific value, the highest score was obtained by the Apodi-Mossoró Estuary with 26 points, followed by the Rosado Cliffs with 25 points, the Rosado Dunes with 24 points, the Conchas River Estuary with 23 points, and Ponta do Mel and the hypersaline desert of Porto do Mangue, which both achieved 22 points (Table 7).

**Table 7.** Assessment of the sites of interest according to Diniz, Araújo, and Chagas for the scientific value.

| | | | | **Scientific Value** | | | | | |
|---|---|---|---|---|---|---|---|---|---|
| **Sites** | **A1** | **A2** | **A3** | **A4** | **A5** | **A6** | **A7** | **Total** | **Evaluation Classes** |
| A—Pedra do Chapéu | 2 | 4 | 2 | 2 | 1 | 1 | 4 | 16 | Medium |
| B—Gado Bravo Beach | 0 | 4 | 0 | 1 | 1 | 1 | 0 | 7 | Low |
| C—Areias Alvas Beach | 0 | 4 | 1 | 2 | 2 | 0 | 0 | 9 | Low |
| D—Barra Beach | 0 | 4 | 1 | 2 | 1 | 0 | 1 | 9 | Low |
| E—Apodi-Mossoró Estuary | 4 | 4 | 4 | 4 | 3 | 4 | 3 | 26 | **High** |
| F—Upanema Beach | 0 | 4 | 3 | 2 | 1 | 3 | 2 | 15 | Medium |
| G—São Cristóvão Beach | 2 | 4 | 3 | 3 | 0 | 4 | 3 | 19 | Medium |
| H—Ponta do Mel Beach | 2 | 4 | 4 | 4 | 1 | 4 | 3 | 22 | **High** |
| I—Porto do Mangue Hypersaline Desert | 4 | 0 | 4 | 2 | 4 | 4 | 4 | 22 | **High** |
| J—Rosado Cliffs | 4 | 4 | 4 | 4 | 3 | 4 | 4 | 25 | **High** |
| K—Rosado Dunes | 4 | 4 | 4 | 4 | 4 | 0 | 4 | 24 | **High** |
| L—Conchas River Estuary | 4 | 4 | 4 | 2 | 4 | 1 | 4 | 23 | **High** |

Legend: A1 (degree of knowledge), A2 (ecodynamics of the environment), A3 (representativeness of geomorphological materials and processes), A4 (diversity of geomorphological aspects—forms and processes), A5 (ecological interest), A6 (paleogeographic value), and A7 (didactic relevance). Highlighted are the sites with the highest value.

The São Cristóvão Beach, Pedra do Chapéu, and Upanema Beach, achieved 19, 16, and 15 points, respectively, scores which were considered medium. The Areias Alvas and Barra beaches scored low (9 points) and the Gado Bravo Beach scored the lowest with 7 points. In general, the low score for these criteria reflected the scarcity of scientific studies involving the areas.

For the aesthetic value, Ponta do Mel, the Rosado Cliffs, the São Cristóvão Beach, and the Rosado Dunes sites were valued with the highest scores of 17, 17, 16, and 16 points, respectively (Figure 12). As shown in Table 8, these were the sites with the highest aesthetic values.

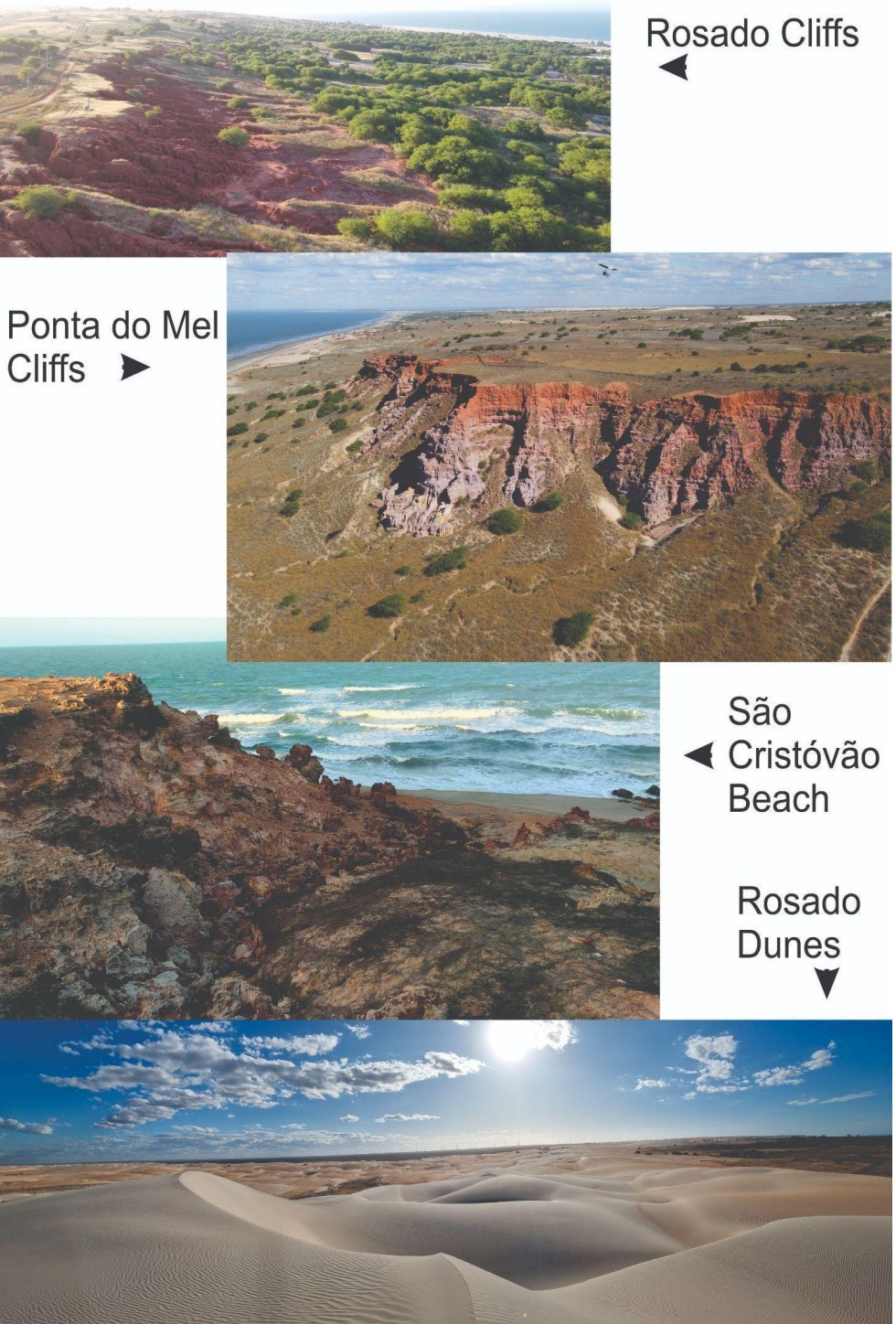

**Figure 12.** Photographs of geomorphosites with high aesthetic values. Source: the authors.

**Table 8.** Evaluation of the sites of interest according to Diniz, Araújo, and Chagas (2021) for the aesthetic value.

| Aesthetic Value | | | | | | | |
|---|---|---|---|---|---|---|---|
| Sites | B1 | B2 | B3 | B4 | B5 | Total | Evaluation Classes |
| A—Pedra do Chapéu | 2 | 2 | 2 | 2 | 4 | 12 | Medium |
| B—Gado Bravo Beach | 0 | 1 | 1 | 0 | 3 | 5 | Low |
| C—Areias Alvas Beach | 2 | 4 | 3 | 1 | 3 | 13 | Medium |
| D—Barra Beach | 2 | 3 | 2 | 2 | 3 | 12 | Medium |
| E—Apodi-Mossoró Estuary | 2 | 2 | 4 | 2 | 3 | 13 | Medium |
| F—Upanema Beach | 3 | 3 | 2 | 2 | 3 | 13 | Medium |
| G—São Cristóvão Beach | 3 | 3 | 4 | 2 | 4 | 16 | **High** |
| H—Ponta do Mel Beach | 3 | 3 | 4 | 3 | 4 | 17 | **High** |
| I—Porto do Mangue Hypersaline Desert | 3 | 4 | 1 | 0 | 4 | 12 | Medium |
| J—Rosado Cliffs | 4 | 4 | 2 | 3 | 4 | 17 | **High** |
| K—Rosado Dunes | 4 | 4 | 1 | 3 | 4 | 16 | **High** |
| L—Conchas River Estuary | 2 | 3 | 2 | 3 | 2 | 12 | Medium |

Legend: B1 (rarity), B2 (integrity), B3 (variety of elements of geodiversity and/or associated themes), B4 (visual quality), and B5 (observation conditions). Highlighted are the sites with the highest value.

The Areias Alvas Beach, Apodi-Mossoró Estuary, and Upanema Beach each achieved 13 points, while Pedra do Chapéu, the Hypersaline Desert of Porto do Mangue, the Barra Beach, and Conchas River Estuary all obtained 12 points, representing a medium score. A very low score of 5 points was only assigned to the Gado Bravo Beach

Pedra do Chapéu, with 16 points, was the most representative site in the touristic value category. The Apodi-Mossoró Estuary and Ponta do Mel obtained a medium value of 15 points, followed by the Upanema Beach with 14 points, the São Cristóvão Beach and the Rosado Dunes with 13 points, the Rosado Cliffs and Barra Beach with 12 points, and the Conchas River Estuary with 11 points (Table 9).

**Table 9.** Evaluation of the sites of interest according to Diniz, Araújo, and Chagas (2021) for the touristic value.

| Touristic Value | | | | | | | |
|---|---|---|---|---|---|---|---|
| Sites | C1 | C2 | C3 | C4 | C5 | Total | Evaluation Classes |
| A—Pedra do Chapéu | 4 | 4 | 3 | 2 | 3 | 16 | **High** |
| B—Gado Bravo Beach | 1 | 4 | 3 | 0 | 2 | 10 | Low |
| C—Areias Alvas Beach | 3 | 2 | 0 | 0 | 1 | 6 | Low |
| D—Barra Beach | 4 | 3 | 2 | 1 | 2 | 12 | Medium |
| E—Apodi-Mossoró Estuary | 4 | 4 | 2 | 2 | 3 | 15 | Medium |
| F—Upanema Beach | 4 | 4 | 2 | 1 | 3 | 14 | Medium |
| G—São Cristóvão Beach | 4 | 3 | 2 | 2 | 2 | 13 | Medium |
| H—Ponta do Mel Beach | 3 | 3 | 2 | 3 | 4 | 15 | Medium |
| I—Porto do Mangue Hypersaline Desert | 4 | 0 | 0 | 0 | 2 | 6 | Low |
| J—Rosado Cliffs | 4 | 3 | 1 | 0 | 4 | 12 | Medium |
| K—Rosado Dunes | 4 | 4 | 1 | 2 | 2 | 13 | Medium |
| L—Conchas River Estuary | 4 | 4 | 1 | 1 | 1 | 11 | Medium |

Legend: C1 (accessibility), C2 (presence of infrastructure), C3 (existence of ongoing use), C4 (scenery), and C5 (tourism category). Highlighted is the site with the highest value.

The sites that achieved low scores for these values were the Gado Bravo Beach, Areias Alvas Beach, and the Porto do Mangue Hypersaline Desert, with scores of 10, 6, and 6, respectively.

The abovementioned scores, as well as the differentiation of the sites against the others, are illustrated in the cartographic representation of Figure 7, following the proposed representation of Reynard et al. [10] which presented the potential geomorphosites according to the values obtained.

## 4. Discussion

In line with the methodology of Reynard [8,9], Ponta do Mel, the hypersaline desert of Porto do Mangue, the Rosado Cliffs, and the Rosado Dunes were considered geomorphosites, totalling four out of the twelve sites evaluated with this method. According to the method proposed by Diniz, Araújo, and Chagas [11], the same sites were considered geomorphosites, alongside three others, namely the Apodi-Mossoró Estuary (high scientific value), the São Cristóvão Beach (high aesthetic value), and the Conchas River Estuary (high scientific value), totalling seven geomorphosites.

In comparison with the results obtained by Diniz, Araújo, and Chagas [11] for Icapuí (Ceará–Brazil), when applying the method, eight sites of interest were quantified, of which seven obtained high values for scientific and/or aesthetic criteria and were thus identified as geomorphosites. Of these, three sites scored high only for the scientific value, two only for the aesthetic value, and two scored high on both criteria (aesthetic and scientific). The aesthetic criterion was relevant to the definition of geomorphosites, as it included two sites that would not be considered in the scientific criterion alone.

The Reynard methodology [8,9] was also used in the work of Diniz, Araújo, and Chagas [11] to quantify the eight sites of interest, four of which scored high for the scientific criterion. This indicates that the authors' proposal was more comprehensive in encompassing the aesthetic dimensions of the sites, including relevant areas that would not reach high values for the scientific criterion.

### 4.1. Geomorphosites Identified Using the Reynard Method

Only four sites were evaluated as geomorphosites using the Reynard method [8,9]. Ponta do Mel obtained a high scientific value with a maximum valuation due to the integrity of the site; all features were preserved at this site, as well as being representative and rare within the context to which it belongs, containing a paleogeographic value. Ponta do Mel also achieved high scores for the additional values, namely the ecological and aesthetic ones.

The Porto do Mangue Hypersaline Desert obtained a high score for scientific value due to its rarity, representativeness, and great integrity. The Rosado Cliffs site also reached maximum points in this category due to its good preservation conditions, its representativeness, rarity in context, and paleogeographic value for telling the history of the Earth.

The Rosado Dunes achieved a high score of 0.81 due to their integrity, as well as their representativeness in the regional context, having a rare occurrence but little paleogeographic representativeness.

### 4.2. Geomorphosites Identified Using the Diniz, Araújo, and Chagas (2021) Method

The geomorphosites Apodi-Mossoró Estuary, Ponta do Mel, Porto do Mangue Hypersaline Desert, Rosado Cliffs, Rosado Dunes, and Conchas River Estuary achieved high scores of scientific value due to the existence of scientific studies (theses, technical studies, articles, academic journals, . . . ) encompassing the areas where they occur. In addition, these geomorphosites can help illustrative geomorphological processes that can be used as didactic resources in different schools, on different academic levels, and for the general public, besides being configured as a strongly unstable environment, susceptible to morphogenesis.

The São Cristóvão Beach, Ponta do Mel, Rosado Cliffs, and Rosado Dunes stood out in terms of their aesthetic value. These sites were distinguished by their geomorphologic importance in relation to the regional context, the good visualisation conditions of the geodiversity elements, and the importance of their occurrence in the regional context. It is

worth mentioning that the use of the aesthetic value as a defining criterion was extremely important for the São Cristóvão Beach to be considered a geomorphosite, considering the scarcity of scientific knowledge regarding the area and the near absence of vegetation within it.

## 5. Comparison of the Quantifications Using the Two Methods

A map summarizing the location of the sites, identifying those that are geomorphosites or geodiversity sites, and their quantification of core values is presented in Figure 13.

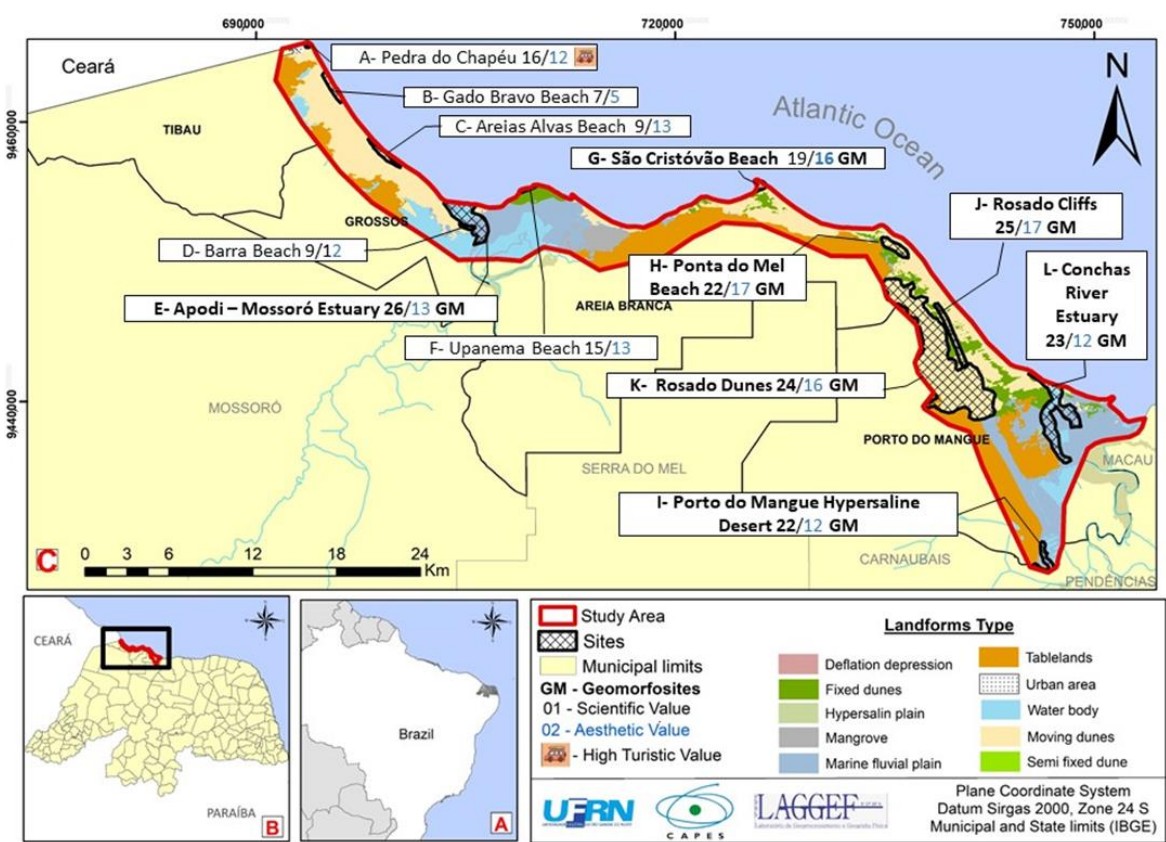

**Figure 13.** (**A**) Location of the state of Rio Grande do Norte in South America. (**B**) Location of the area in the state of Rio Grande do Norte and (**C**) Synthesis map of the geomorphological sites of interest in Costa Branca—Rio Grande do Norte. Source: the authors.

Although the methodology of Reynard [8–10] was able to cover the geomorphological aspects of the sites, some nuances of it were expanded on in the methodological approach of Diniz, Araújo, and Chagas [11]. For example, the aesthetic element, which included the verticality dimension, colour contrasts, observation conditions, variety of geodiversity elements, rarity, and integrity was relevant for the identification of areas of geomorphological interest with unique aspects; these included the São Cristóvão Beach, which could be considered a geomorphosite only due to its aesthetic value. This beach yielded a high paleographic value for being a small repercussion on the coast of the Serra do Mel dome uplift, but had medium representativeness and a diminished integrity due to the presence of a small village in the area which included the community cemetery that was built on the cliffs.

The São Cristóvão Beach cemetery revealed a very curious aesthetic aspect, as its wall can be used as an excellent viewpoint of the sunset over the area's inlet; the touristic sector could use expressions such "the cemetery has the most beautiful view on the Brazilian coast" (Figure 14).

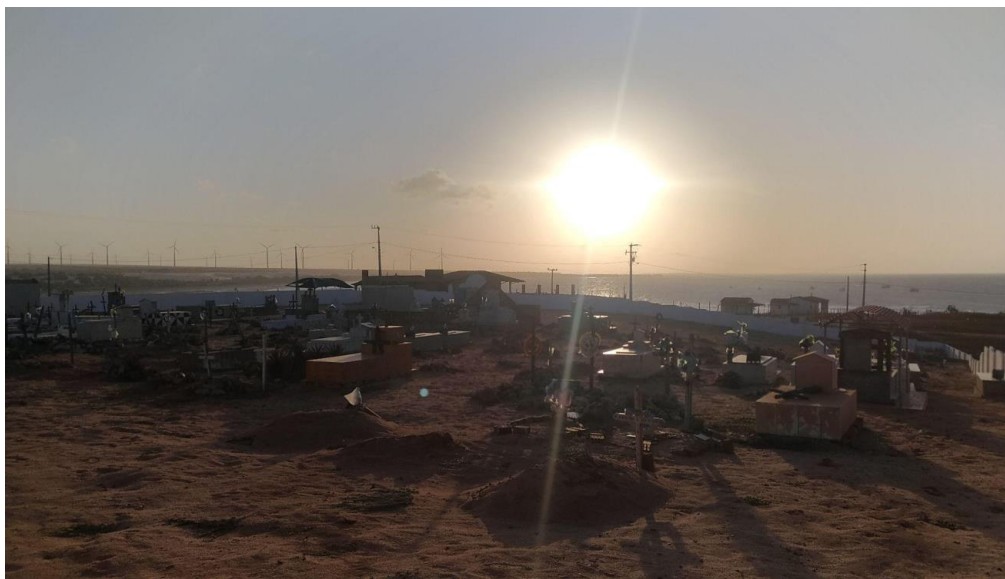

**Figure 14.** Sunset view from the São Cristóvão Beach cemetery. Source: the authors.

It is understood that it is necessary to quantify these aesthetic factors, in order to apprehend and highlight specificities that would not be considered in an appropriate manner solely through scientific elements. Thus, according to the results obtained, the inclusion of aesthetic criteria was relevant for the identification and definition of areas of geomorphological interest and geoconservation.

The Conchas River and Apodi-Mossoró estuaries were considered to have a high scientific value according to the Diniz, Araújo, and Chagas [11] method which captured relevant criteria for estuarine environments such as the degree of scientific knowledge about the area (numerous scientific papers have been published about these estuaries) and its ecological interest. Both these values would be high in areas of great biodiversity such as estuaries, where this biodiversity is conditioned by the geomorphology of the area.

The abovementioned estuaries and the São Cristóvão Beach were considered geomorphosites when using the method of Diniz, Araújo, and Chagas [11], alongside Ponta do Mel, the Rosado Dunes, Rosado Cliffs, and Hypersaline Desert. Quantification using the method of Reynard [8–10] only classified the latter four sites as ones of interest.

## 6. Conclusions

The quantitative valuation of natural elements is an important part of the process of appropriating the potential use and conservation of areas of abiotic diversity. However, this practice, particularly within geodiversity studies, is often subjective and developed according to the researcher's attribution.

Thus, the results obtained with the application of the quantification methods for the coastal areas of Tibau, Grossos, Areia Branca, and Porto do Mangue/RN showed that aesthetic criteria should be included in the identification and definition of areas of potential geomorphological interest. We observed that some areas scoring highly on aesthetic criteria had little scientific value, due to the lack of research covering these areas. This is a common reality in Brazil, due to the large extent of the territory and inaccessibility to some areas. This corroborates the view that the aesthetic value is representative of the society and therefore should be included in the conservation actions.

Among the sites quantified, we argue that there was a greater need for actions to promote geoconservation in those with high scores with aesthetic and scientific values. In view of the fact that they corresponded to sites with aesthetic spectacularity and/or morphological singularity, with development of studies, there is need for the preservation of their features, practical initiatives that promote the participation of the local community, governments, and private initiatives within the context of geoconservation.

The Costa Branca presents the potential for the creation of a geopark project, since it presents sites of high scientific and aesthetic value. Some of these sites were considered unique places for the understanding of Quaternary tectonics in the passive margin of the Atlantic in South America. These were distinguished in this case by the cliffs of Ponta do Mel, Rosado, and São Cristóvão Beach. These areas were repercussions on the coast of Quaternary tectonic movements; some of these movements dated back to the end of the Quaternary) as in those of the Rosado Cliffs.

Conversations were initiated with the government of the state of Rio Grande do Norte, with municipalities and a civil society that made up for Polo Costa Branca for the beginning of the project of the Geopark for the area; the results of this article will be fundamental to this future proposal.

**Author Contributions:** Conceptualization, M.T.M.D.; Methodology, M.T.M.D. and M.L.d.O.T.; Validation, M.L.d.O.T.; Investigation, M.L.d.O.T. and F.E.B.d.S.; Data curation, F.E.B.d.S.; Writing—original draft, M.L.d.O.T. and F.E.B.d.S.; Writing—review & editing, M.T.M.D.; Supervision, M.T.M.D. All authors have read and agreed to the published version of the manuscript.

**Funding:** This research was funded by the Coordination for the Improvement of Higher Education Personnel (CAPES) and Research Support Foundation of Rio Grande do Norte (FAPERN) and The APC was funded by Federal University of Rio Grande do Norte.

**Data Availability Statement:** Not applicable.

**Acknowledgments:** The authors gratefully acknowledge the National Council for Scientific and Technological Development (CNPq) and the Research Support Foundation of Rio Grande do Norte (FAPERN) for funding the field activities. The Research Productivity grant provided funding for the first author and the doctoral scholarship from the Coordination for the Improvement of Higher Education Personnel (CAPES) supported the second author.

**Conflicts of Interest:** The authors declare no conflict of interest.

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
