# Peer review of "Assessment of the Geomorphological Heritage of the Costa Branca Area, a Potential Geopark in Brazil"

_resources, doi:10.3390/resources12010013_

Round 1

Reviewer 1 Report

Dear Authors,

Your manuscript sounds good but it need to be improved. The Methodology, the results and discussion sections must be improved. My comments and suggestions are found in the Reviewed version of the manuscript.

Best regards.

Author Response

Letter - Reviewer 1

Dear reviewer, the following changes have been made:

- Line 2 - The concept was better worked out in the text.

- Line 42 - The authors have been listed.

- Line 58 - The change was made.

- Line 60 - 61 - We chose to insert a paragraph dealing with the theme of geoparks, which had not been included.

- Line 80 - The changes were made in the map.

- Line 102 - The reference was added.

- Line 135 - The names of the figures were inserted.

- Line 142 - The paragraph has been added.

- Line 160 - The representation mode has been changed.

- Line 173 - Reference has been added.

- Line 206 - Rosado Cliffs captured with a drone.

- Line 207 - 213 - The paragraph has been changed to make it easier to understand.

- Line 256 - We have used the term "Geomorphological heritage".

- Line 402 - The section was improved, we added the comparison of the results obtained in the application of the same methodology used ( Diniz, Araújo e Chagas, 2022) in another study area (Litoral de Icapuí - Ceará/ Brasil) by the same authors.

Reviewer 2 Report

The authors propose a very interesting discussion of methods for geodiversity research, introducing in a very distinct way the aesthetic value in the process of geoheritage evaluation. From their perspective and point of view, the method presents no weakness. We can argue if it is really interesting to use aesthetic value for the purpose of geoheritage definition. Geoheritage is considered as important areas that deserve to be preserved for future generations. Should beauty be included in this consideration? That is a big question! The authors argue that yes, it should, but in reality, it would maybe increase the number of geomorphosites/preserved areas to a total incompatible to modern life. If we work with a higher number of sites of geomorphic interest, we could also do geotourism and geoconservation, without "inflation" of geoheritage sites, needed to be preserved to the humankind. Nevertheless, though these aspects, the subject merits to reach a larger number of researchers, in order to promote discussion which could perfectionate the methodologies of research in geodiversity, and for this reason, we are favorable to its publication. However, the paper needs a major revision. The introductory parts, the ones that presents the sites, are full of incorrections and mistakes that need to be addressed. The language also needs attention. We list below the points that took our attention, to get the consideration of the authors:

Line 28 – the authors do not consider also the climate as an element of geodiversity?

Line 39 – instead of mankind, it is preferable to use the word humankind

Line 57 – It is strange to say that the geodiversity has objectives. Geodiversity is an element of nature, has no objectives to exist. The authors may want to say geoconservation, directly linked to geodiversity, as geotourism and geoeducation.

Line 61 – The preservation of only “cute animals” is a distortion of the principles of nature conservation and should not be transported as a value to geodiversity discussions.

Line 70 – The limits of Brazilian Atlantic margin are with French Guyana and Uruguay, not with Guyana and Argentina

Line 72 – Many authors report Quaternary activity at the coast of Bahia (Martin, Suguio, Landin, Bittencourt, and other authors who work with Quaternary sea level rise).

Line 73 – What the authors mean with the phrase “ is fixed in the Brazilian coast”?

Line 78 – Figure 1: the areas with hatch (hachuras) are not identified in the legend. The legend says the yellow represents the limit of municipality, but all the area is in yellow. The caption says that the source is the author, but the paper has three authors.

Lines 92-94  – The Potiguar Basin evolved from an intracontinental rift, so it is not really a marginal sedimentary basin. It was formed before the opening and separation of Gondwana.

Line 111-112 – The information about the Barreiras Formation has to be reframed, because it can not be cliffed and tabular at the same time (it is tabular with sea limits cliffed, right?).

Lines 117-118 – The explanation of the origins of Serra do Mel is confused. It talks about extension and compression at the same. It is a fact that it can happens some times in the borders of the structures, but it has to be better explained. Besides, the text informs that Serra do Mel is quaternary, but sets the compression in the Cretaceous, in the opening of the equatorial margin.

Line 160 – The Barreiras Formation presents cliffs 7 to 15 m high all over the Brazilian Northeast coastal area. How can an altitude of 10 m indicate uplifting of the coast?

Lines 162-167 – Repetition of text

Line 169 – Figure 4 – The caption says “autors” instead of “authors” (as in almost all figures in the sequence).

Line 177 – Figure 6 – The Digital Elevation Model indicates that Serra do Mel has an altitude of 250 m, but the lines 119 and 120 says “the geomorphologic expression of the dome's uplift is exteriorized in the cliffs, which reach an altitude of about 120 m in Ponta do Mel”. So, it is confusing.

Line 183 – It is hard to understand how a fluviomarine plain can have impermeable soils.

Line 204 – Figure 8 :  The legend says “pos-barrier”, and it is “pos-Barreiras”. Fault is written Falt.

Line 210 – Where is the source for this information of the occurrence of a transgression at 2100 years ago?

Line 214 – Figure 9 : it is not clear what are the transects and the letters and numbers attached to them.

Line 223 –There is no other occurrence of reddish dunes in Brazil? The authors may rephrase to mobile, active dunes, because deposits of reddish sand in Ceara and Piaui States are considered as palaeodunes. And there are stretches of reddish mobile, active dunes in Ceara State, maybe not only so extensive.

Line 228 – Carcinicultura in English is known as “Shrimp farming”.

Line 231 Figure 10. It needs the position of north (orientation).

Line 493 – Passive margin, and not Pacific Margin.

Line 495 – Holocene is part of the Quaternary, so the phrase is strange.

Author Response

Dear reviewer,

Please find the attachment for seeing revision.

Round 2

Reviewer 2 Report

After the corrections, I think the text is good.